# Perceptions Matter! Active Physical Recreation Participation of Children with High and Low Actual and Perceived Physical Competence

**DOI:** 10.3390/ijerph21091129

**Published:** 2024-08-27

**Authors:** Stephanie C. Field, John T. Foley, Patti-Jean Naylor, Viviene A. Temple

**Affiliations:** 1School of Exercise Science, Physical, and Health Education, University of Victoria, Victoria, BC V8P 5C2, Canada; pjnaylor@uvic.ca (P.-J.N.); vtemple@uvic.ca (V.A.T.); 2Department of Physical Education, State University of New York (SUNY) at Cortland, Cortland, NY 13045, USA; john.foley@cortland.edu

**Keywords:** motor development, physical activity, accuracy

## Abstract

Emerging evidence suggests that the accuracy of a child’s perceived physical competence (PPC) impacts participation in physical activity. We examined differences in active physical recreation (APR) participation based on clusters of high and low motor competence and perceptions from grades 3–5. Participants were a longitudinal sample (*n* = 155; 85 girls) of grade three, four, and five children. Fundamental motor skills (FMSs) were assessed using the TGMD-2, perceptions were assessed using the Self-Perception Profile for Children, and APR was measured using the Children’s Assessment of Participation and Enjoyment. K-means cluster analysis was used to create four clusters per grade based on participants’ FMSs and PPC *z*-scores. Differences in FMSs, PPC, and APR with cluster group as a factor were examined using a series of factorial ANOVAs. In each grade, participants in the high–high cluster participated in significantly more APR than those in the low–low cluster (*p* = 0.002 in grades three and four; *p* < 0.001 in grade five). Among the less accurate clusters (e.g., low FMSs with high PPC) was a trend toward positive FMSs growth among the children with higher perceptions. Results show that combinations of FMSs and PPC influence patterns of engagement or disengagement in active physical recreation persistently across middle childhood.

## 1. Introduction

There is a widely accepted view that factors that influence participation in physical activity are complex [1]. There are many individual (e.g., physical abilities) and environmental (e.g., socioeconomic status) factors that contribute to participation during middle childhood [2,3]. Two individual factors that have been widely researched are motor skill competence and levels of perceived physical competence [4,5,6]. Children who demonstrate high motor skill competence, particularly object control proficiency, are more likely to participate in physical activities compared with those who demonstrate low motor skill competence [5,7]. Two mechanisms are likely at play. The first is that children with more developed motor skills have greater capacity to engage (e.g., catching skills are useful for playing Frisbee), and the second is that higher motor competence contributes to more positive self-perceptions [6]. These individual factors (i.e., motor skills and perceptions) are often examined through a sex-based lens [8]. It is consistently reported that boys have higher object control skill scores than girls (e.g., [9,10,11]), and there are minimal or no sex-based differences in locomotor skills [12,13,14]. Further, boys’ levels of perceptions of their abilities in middle childhood tend to be higher than girls’ [10,13,15,16,17], which may be a reflection of boys’ higher object control competence [10,13]. This trend is supported by several cross-sectional studies examining perceptions of physical competence in middle childhood (e.g., [10,13,17,18]) as well as longitudinal studies [16,19,20]. 

One mechanism receiving increased attention in physical activity participation research is the accuracy of a child’s perceived physical competence [4,8,16,21]. As children move through middle childhood (~6–10 years) [22], the accuracy of their perceived physical competence should improve as a result of increasing motor skill competence and a decrease in the inflated perceived physical competence children exhibit in early childhood [6,23,24,25]. Developmental experts theorize that children should be able to develop accurate self-appraisals by seven years of age [23,24]; however, recent literature suggests the accuracy of perceived physical competence occurs later [15,16,19]. The extant literature reveals that the combination of motor skill competence and perceptions of physical competence influences participation in physical activities [15], and that most children have accurate perceptions of their physical abilities by grade three (~8 years of age) [16]. In combination, high motor skill levels and high perceptions of competence are likely to contribute to a positive spiral of engagement in physical activities, providing both the tools and confidence to participate [6]. 

Understanding the accuracy of perceptions of physical competence and how it influences participation in physical activity is particularly important in the case of children who underestimate their abilities. Children who underestimate their abilities have been shown to have higher anxiety, lower motivation, and are less aware of the locus of control of their performance in physical activity settings relative to their peers, all of which may contribute to withdrawal from physical activity [26,27]. Further, these underestimators may have the skills required to participate in physical activity; however, due to low levels of perceived physical competence, they may choose to withdraw from activity [6,27]. Having low perceptions of competence may create a self-fulfilling prophecy, where children who perceive their skills to be low may end up with low motor competence as a result of not participating in physical activities that foster skill development [6]; however, there is not yet sufficient evidence to determine this [4]. This premise is partially supported by longitudinal evidence found in our previous work, that children with high motor skills relative to their peers in grade two, but who underestimated their physical abilities (i.e., had low perceptions of competence), had low motor competence relative to their peers by grade four [16]. De Meester and colleagues [21] furthered this line of research by examining the relationships between perceived physical competence and motor skill levels on participation in physical activity among 9-year-old children. These authors found that children with low perceived physical competence in combination with low skill levels relative to their peers had lower physical activity participation. Conversely, a child’s overestimation of their skills may encourage participation. This thought is supported by recent findings that 6- and 10-year-old children had inflated perceptions of object control proficiency, and those inflated perceptions were a significant contributor to physical activity participation [28]. Therefore, the aim of this study was to examine differences in active physical recreation participation based on clusters of high and low motor competence and high and low perceptions of physical competence from grade three to grade five.

## 2. Materials and Methods

This study used a longitudinal research design with data that were collected as part of a larger study that took place from 2010 to 2017 in Victoria, British Columbia. The University of Victoria Human Research Ethics Board as well as the participating school district granted approval (protocol number 10-246). 

### 2.1. Participants

Children attending grade three at one of eight participating elementary schools in Victoria, British Columbia, were invited to participate. Participants were recruited from two cohorts of grade three students (2013–2014 and 2014–2015) and subsequently tracked into grades four and five. Parents or guardians provided written informed consent at the beginning of each data collection year, and children provided written assent. Consent was obtained for 428 children to participate, and of those children, 155 had complete data for grades three, four, and five. The mean age of participants in grade three was 8.8 years (*SD* = 4.0 months) and 55% were female. It is important to note that this study utilizes sex-based analysis and not gender-based analysis. At the time of data collection for this study, demographic questions included the participant’s sex, not gender; therefore, it would be inappropriate for the authors to comment on participants’ gender identities as this information was not collected. 

### 2.2. Measures

Motor skills were assessed using the Test of Gross Motor Development, Second Edition (TGMD-2) [29], and perceived physical competence was assessed using the Self-Perception Profile for Children [30,31]. Active physical recreation was assessed using the Children’s Assessment of Participation and Enjoyment (CAPE) [32]. These measures and the accompanying procedures are outlined below.

### 2.3. Fundamental Motor Skills

The TGMD-2 is a 12-item, norm- and criterion-referenced, developmental fundamental motor skills test that includes two subscales: locomotor skills (run, gallop, slide, hop, jump, and leap) and object control skills (dribble, catch, strike, throw, roll, and kick). Each item on the locomotor or object control skills subscale is individually assessed and is comprised of 3–5 components (e.g., when a child runs, their “arms move in opposition to legs, elbows bent”). Every time a child performs a skill component according to the TGMD-2 criteria, they are awarded a ‘1’. Children receive a ‘0’ if the criterion is not met. The raw score range for each subscale is 0–48, with a combined locomotor and object control raw score range of 0–96. Raw motor skills scores were used in lieu of standard scores in this study, as TGMD-2 standard scores were generated from a normative sample located outside Canada and are representative of children from 20 years ago. The TGMD-2 is validated for children 3.0–10.11 years of age. Test–retest reliability is reported at 0.94 (locomotor) and 0.96 (object control) for 6–8-year-old children and 0.86 (locomotor) and 0.84 (object control) for 9–10-year-old children [29]. Criterion-prediction validity was established through moderate to strong partial correlations (locomotor = 0.63; object control = 0.41) and the goodness-of-fit index was reported as 0.96 [29]. The Percent Agreement Method [33] was used to estimate inter- and intra-rater reliability for the TGMD-2 skills. Inter-rater reliability between the primary investigator and a second trained research assistant was 0.88, and intra-rater reliability was 0.98.

### 2.4. Perceived Physical Competence

The Self-Perception Profile for Children [31] is a 36-item self-report survey for children in grades 3–8 that contains six subscales: scholastic competence, social competence, athletic competence, physical appearance, behavioral conduct, and global self-worth [31]. The athletic competence subscale was used in this study. Each subscale consists of six items, and the score range for each item is 1–4, resulting in a total perceived athletic competence score range of 6–24. Each subscale item consists of a positive and negative statement; one of a child who is not proficient in the task (e.g., some kids wish they could be a lot better at sports), and one of a child who is proficient in the task (e.g., other kids feel they are good enough at sports). The child selects which statement is most like them and if their answer is ‘really true’ (e.g., it is really true that they could be a lot better at sports) or ‘sort of true’ (e.g., it is sort of true that they could be a lot better at sports). Positive statements are associated with higher scores (e.g., 3 or 4) and negative statements with lower scores (e.g., 1 or 2) depending on the selection of ‘really true’ or ‘sort of true’. Internal consistency reliability for grade 3–6 children ranged from 0.76 to 0.85 and exploratory and confirmatory factor analysis from 0.72 to 0.90 [31]. There are many phrases used in the developmental literature to discuss the domain of perceived competence: perceived physical competence, perceived motor competence, perceived athletic competence, and so forth. We used the phrase ‘perceived physical competence’ to refer to this domain, as it is commonly used in related literature. 

### 2.5. Active Physical Recreation

Active physical recreation (i.e., physical activity) participation was measured using the CAPE [32]. The CAPE is a 55-item survey for individuals 6–21 years of age that assesses five dimensions of participation in voluntary recreation and leisure activities completed within the previous four months: diversity (the number of activities done), intensity (how often an activity is done), with whom, where, and enjoyment [32]. CAPE items are divided into nine activity categories, including informal active physical recreation (11 items) and formal organized sports (6 items). Organized sports included practice times, coaching sessions, and competitions. Each item within these two activity categories (a total of 17 items) requires physical exertion [32] and the term active physical recreation is used throughout this study to represent both categories to remain consistent with CAPE terminology. Internal consistency of the CAPE is reported at 0.35–0.42 (formal activities) and 0.76–0.77 (informal activities) [32]. Test–retest correlations for activity intensity range from 0.72 to 0.81 [32]. Content validity for the CAPE was established through a thorough literature review of participation, expert review, and pilot work [32].

CAPE intensity scores (i.e., how often a child reported participating in an activity) were calculated for boys and girls in grades 3, 4, and 5. As per the CAPE manual [32], to calculate an intensity score, the sum of intensity ratings (see Table 1) is divided by the number of possible items (i.e., 17 items). It is important to note that the CAPE category of intensity is operationally defined by the CAPE manual as how often a child participates in an activity; essentially, this category is synonymous with frequency of participation as opposed to energy expenditure (e.g., light, moderate, vigorous). Through the remainder of this study, we will continue to refer to intensity to discuss how often children participate in activities, to remain consistent with CAPE labels. 

### 2.6. Procedures

Consistent with the respective assessment tool manuals, research assistants were trained annually by the project coordinator and lead investigator in data collection procedures for fundamental motor skills, perceived physical competence, and active physical recreation [29,30,31,32]. 

The TGMD-2 was administered over two, 30-min physical education classes. Occasionally, physical education classes were one hour long, which allowed for data collection in one session. The project coordinator divided children into four groups of students who subsequently rotated through four stations in the school gymnasium. All children participated in data collection, yet only children who consented to participate were video recorded (Sony Handycam HDR-CX240, Toronto, ON, Canada). Digital video was downloaded onto a secure server at the University of Victoria and scored by a researcher trained in scoring procedures consistent with the TGMD-2 manual [29]. 

Perceptions of physical competence and CAPE questionnaires were completed one-on-one with a trained research assistant in a quiet setting (e.g., school library, multi-purpose room) during school hours. Completion of both questionnaires took between 20 and 40 min. Depending on the time allowance, both questionnaires were completed on the same day; otherwise, the project coordinator would arrange for a second visit to the school. Prior to completing the perceptions questionnaire, children were reminded that there were no right or wrong answers and to respond only with what felt true to them. After administration of the first few questions, some participants expressed the desire to complete the Self-Perception Profile for Children independently. In these cases, the administrator remained seated with the participant to answer any clarifying questions and to ensure correct completion of the questionnaire. 

Prior to completing the CAPE, children were instructed to only respond for activities they had voluntarily participated in outside of class time within the previous four months; as per the CAPE manual, this is a timeframe children can reasonably recall [32]. They were also reminded that there were no right or wrong answers. As per the CAPE administration manual and materials, children were provided with a picture binder that included one drawing that corresponded to each CAPE item in order to assist the children’s understanding [32]. Children followed along with the picture binder while the research assistant administered the questionnaire. In some cases, children needed encouragement to provide responses, and as per the CAPE administration protocol [32], probing for more information was occasionally necessary and completed by asking children for clarification or using the picture binder as an aid (e.g., providing an example such as ‘tag’ to help a child understand if they had ‘played games’ in the last four months); however, research assistants were careful not to lead the child’s response. 

### 2.7. Data Treatment and Analyses

The findings from our previous work on the accuracy of perceived physical competence in middle childhood [16] served as a preliminary analysis for this study. In that study, we examined the trajectory of children’s accuracy of perceived competence longitudinally from grade two to grade four and found that the majority of participants had developed accurate perceptions of physical competence by grade three (for full methods and results, please see [16]), although some children did remain inaccurate in their self-perceptions. Participants in the current study are from the same longitudinal sample of children and were selected beginning in grade three based on the findings of the preliminary analysis. 

K-means cluster analysis [34] was used to create four clusters per grade based on participants’ grade three, four, and five motor skills and perceptions of physical competence scores. K-means cluster analysis is a widely used clustering algorithm that involves the user indicating the number of desired clusters (i.e., four), and specifying ‘K’, the initial centroid, of each cluster [35]. In this study, participants’ motor skills and perceptions raw scores were converted into *z*-scores to facilitate K-means cluster analysis. The partitional algorithm of K-means cluster analysis avoids overlapping clusters [35]. The four cluster groups were low motor skills–low perceptions (low–low); high motor skills–high perceptions (high–high); low motor skills–high perceptions (low–high); and high motor skills–low perceptions (high–low). Descriptive statistics for the four clusters were computed for motor skills, perceptions, and active physical recreation for children in each grade. It is important to note that the composition (i.e., *n*) within the clusters differed in each grade (see Table 2) as children’s motor skills and perception levels changed over time. For example, a child that may have been in the low–low group in grade three may have increased their motor skills competence, and therefore be in the high–low group in grade four. A series of three 2 × 4 factorial ANOVAs (with sex and cluster group as factors) were run for each grade to look for differences in (1) motor skills, (2) perceptions of physical competence, and (3) active physical recreation. In total, nine factorial ANOVAs were run. Post hoc tests using Bonferroni correction were run for each factorial ANOVA. All analyses were computed in IBM SPSS for Windows, Version 26.0 (NY, USA) and significance was set at *p* < 0.05.

## 3. Results

Descriptive statistics for motor skills, perceptions, and active physical recreation for each cluster are presented in Table 2, and for boys and girls separately in Table 3. Results from a series of 2 × 4 factorial ANOVAs revealed a significant main effect of cluster on motor skills in grade three (*F*(3, 147) = 65.88, η_p_^2^ = 0.573, *p* < 0.001), grade four (*F*(3, 147) = 100.93, η_p_^2^ = 0.673, *p* < 0.001), and grade five (*F*(3, 147) = 137.68, *p* < 0.001, η_p_^2^ = 0.738), with participants in groups characterized as having high motor skills (i.e., high–high and high–low) outperforming those characterized as having low motor skills (i.e., low–low and low–high) with an approximate mean score difference of between 15 and 19 skill components on the TGMD-2, in respective years. Bonferroni post hoc comparisons of motor skills between clusters are reported in Table 2. A main effect of sex on motor skills was present in grade three (*F*(1, 147) = 19.95, *p* < 0.001, η_p_^2^ = 0.120,), grade four (*F*(1, 147) = 57.06, *p* < 0.001, η_p_^2^ = 0.280), and grade five (*F*(1, 147) = 37.43, *p* < 0.001, η_p_^2^ = 0.203,), with girls demonstrating lower motor skills than boys in each grade (see Table 3). There was a significant interaction effect between sex and cluster for grade three (*F*(1, 147) = 3.75, *p* = 0.012, η_p_^2^ = 0.071), but not for grade four (*F*(3, 147) = 0.67, *p* = 0.574, η_p_^2^ = 0.013) or grade five (*F*(3, 147) = 1.14, *p* = 0.334, η_p_^2^ = 0.023).

A significant main effect of the cluster on perceptions of physical competence was present in grade three (*F*(3, 147) = 149.08, *p* < 0.001, η_p_^2^ = 0.753), grade four (*F*(3, 147) = 146.32, *p* = 0.001, η_p_^2^ = 0.749,), and grade five (*F*(3, 147) = 108.96, *p* < 0.001, η_p_^2^ = 0.690), with those in the low–low group reporting a score of ~7–12 points less on the Self-Perception Profile than those in the high–high group. See Table 2 for Bonferroni post hoc comparison results. There was a significant effect of sex on perceptions in grade three (*F*(1, 147) = 31.25, *p* < 0.001, η_p_^2^ = 0.175), grade four (*F*(1, 147) = 11.54, *p* = 0.001, η_p_^2^ = 0.073), and grade five (*F*(1, 147) = 8.61, *p* = 0.004, η_p_^2^ = 0.055), with boys demonstrating higher perceptions in each grade compared to the girls (see Table 3). There were no significant interactions between sex and cluster for grade three (*F*(3, 147) = 1.45, *p* = 0.213, η_p_^2^ = 0.029), grade four (*F*(3, 147) = 0.27, *p* = 0.847, η_p_^2^ = 0.005), or grade five (*F*(3, 147) = 0.36, *p* = 0.782, η_p_^2^ = 0.01).

The main effect of the cluster on participation in active physical recreation was significant in grade three (*F*(3, 147) = 6.20, *p* = 0.001, η_p_^2^ = 0.112), grade four (*F*(3, 147) = 5.29, *p* = 0.002, η_p_^2^ = 0.097), and grade five (*F*(3, 147) = 8.30, *p* < 0.001, η_p_^2^ = 0.145). In each grade, participants in the high–high cluster participated in significantly more active physical recreation than those in the low–low cluster. Bonferroni post hoc comparison results for differences in active physical recreation between the clusters are presented in Table 2. There was no significant effect of sex on active physical recreation participation in grade three (*F*(1, 147) = 0.89, *p* = 0.348, η_p_^2^ = 0.006), grade four (*F*(1, 147) = 2.01, *p* = 0.159, η_p_^2^ = 0.013), or grade five (*F*(1, 147) = 2.04, *p* = 0.155, η_p_^2^ = 0.014). There was also no significant interaction between sex and cluster for active physical recreation participation in grade three (*F*(3, 147) = 1.19, *p* = 0.317, η_p_^2^ = 0.024), grade four (*F*(3, 147) = 1.44, *p* = 0.234, η_p_^2^ = 0.028), or grade five (*F*(3, 147) = 1.95, *p* = 0.124, η_p_^2^ = 0.038).

## 4. Discussion

The findings from this study revealed that the composition of a child’s motor skills proficiency and perceptions of physical competence impacted participation patterns and motor skill development in middle childhood. Accuracy, or inaccuracy, emerged as a factor that influenced participation in active physical recreation in each grade. Overestimators, those with low motor skills and high perceptions, appeared to benefit from their inaccurately high perceptions and had increased participation and skill improvement in each grade. In contrast, underestimators, those with high motor skills and low perceptions, had a stable participation rate and little or no skill development in each grade. These findings suggest that perceptions of physical competence are discriminating and may positively or negatively impact physical activity engagement. The following discussion will highlight key findings and address practical and theoretical implications. 

### 4.1. Accuracy and Developmental Mechanisms

The children in the high–high group in grades three, four, and five participated in significantly more active physical recreation than those in the low–low group in those grades. These findings support the hypothesized relationships depicted in the model of developmental mechanisms influencing the physical activity trajectory of children [6], and De Meester and colleagues’ [15] findings show that children with high motor skills and high perceptions participated in more physical activity. Our findings build on those of De Meester and colleagues by not only including the level of perceptions (low or high) but also by including a preliminary analysis of accuracy in our recent work [16]. The findings from the preliminary analysis allow further interpretation of the results from the high–high and low–low groups by showing that participants in those groups had accurate perceptions, and that those accurate perceptions were impacting participation. These data show that the pattern of engagement or disengagement based on a combination of motor skill competence and perceptions of physical competence persists across middle childhood. Stodden and colleagues’ [6] original model depicted reciprocal relationships between perceived motor competence and physical activity and between perceived motor competence and motor competence in middle childhood and our results support this. Moreover, these authors noted “middle childhood marks the beginning of a period of vulnerability during which children who have lower actual motor skill competence will, correspondingly, demonstrate lower perceived motor skill competence and are less physically active. That is, they will opt out of physical activity because (a) they understand they are not as competent as peers…” ([6], p. 296).

The phrase “they understand they are not as competent as peers” has particular salience for this study. As the children in the low–low cluster were accurate in their perceptions of their abilities, they understood their competence was lower than that of their peers. In every grade, children in the low–low cluster participated in significantly less active physical recreation than in any other cluster. This outcome is consistent with what Stodden and colleagues [6] conceptualized as a compounding spiral of disengagement. With poor motor skills leading to low perceptions of competence (or vice versa), this paves the way to low engagement with, or withdrawal from, physical activities. Conversely, children in the high–high cluster had more developed motor skills, more positive perceptions of their abilities, and were more active. 

### 4.2. Higher Perceptions of Physical Competence Clusters

We cannot directly compare change over time in motor skills, perceptions, and participation within the clusters because the composition of the clusters changed from grade to grade, as indicated in Table 2. However, it is interesting to note that both clusters with higher perceptions of competence (high–high and low–high) appeared to have substantial positive growth in their motor competence from grade three to five; whereas, in the two clusters of children with low perceptions of physical competence (low–low and high–low), there was no development of motor competence. In grade five, the cluster of children classified as having low motor competence with high perceptions of competence had significantly and substantively better motor competence than the children in the low–low cluster did, intimating that having positive perceptions across middle childhood is beneficial to the development of motor competence even among lower-skilled children. These findings are in contrast to a theoretical perspective put forth by Susan Harter [23], in which she argues that children who overestimate their abilities tend to avoid challenging themselves, which consequently inhibits the development of skills and problem-solving abilities. The data in this study can neither directly confirm nor refute how children’s choices are influenced by their positive perceptions. However, our findings warrant further investigation as this information could inform future studies (e.g., how perceptions of competence are examined in physical activity research) and guide practitioners (e.g., educators, coaches) in their delivery of physical activity programs.

### 4.3. Lower Perceptions of Physical Competence Clusters 

Two clusters of children in this study had relatively low perceptions of their physical abilities, the low–low and high–low clusters. As mentioned earlier, the low–low cluster had relatively poor motor competence and they knew it. They also had the lowest levels of active physical recreation participation. To assist this group of children to enter into positive spirals of engagement, creating a positive motivational physical activity climate is recommended [36]. According to Stuntz and Weiss [36], adults (e.g., parents, coaches, teachers) can create a positive motivational climate by providing children with choices, fostering the development of positive and supportive relationships between peers, focusing on feelings of individual competence (e.g., environments free from peer comparison), and by planning activities that are enjoyable for the learner. The combination of these efforts within a motivational environment can have a “dramatic influence on youths’ physical activity motivation and behaviors” ([36], p. 433). 

The second cluster with lower perceptions of physical competence was the high–low cluster. Children in this cluster have relatively high motor competence but low perceptions of their abilities. As can be seen in Table 2, there was no significant difference in motor skills in grade three between the high–high and high–low clusters, but in grades four and 5, the high–high clusters had significantly and meaningfully higher motor skills scores. This apparent stagnation in motor skills development may partly be explained by the competence–motivation theory [23], which posits that children are less motivated to participate in activities if they feel they do not have the skills to succeed. The lack of participation, in turn, suspends the development of motor skills, and so on [6]. 

### 4.4. Sex-Based Differences in Active Physical Recreation 

Although boys in this study demonstrated higher motor proficiency and more positive self-appraisals than girls, there were no significant sex-based differences in active physical recreation. Reports on physical activity levels of boys and girls in childhood (~6–11 years old) indicate that boys are more active than girls [37,38,39]. It is important to note, however, that these differences are often attributed to the intensity of activity as measured by accelerometry [37,38,39], as opposed to frequency or volume of participation. Boys tend to engage in more moderate, and moderate-to-vigorous, physical activity [38], whereas levels of light physical activity are similar between boys and girls [40]. In the current study, participation scores were generated from the frequency of participation for the listed CAPE items and more closely represent the volume of activity a child is doing (e.g., how often) as opposed to the intensity (e.g., moderate or vigorous). Additionally, a subjective measure of active physical recreation participation was used in this study. Other researchers in the field (e.g., [15,28]) have used accelerometry in recent cross-sectional studies. Including an objective measure of participation in longitudinal research would add strength to this area of developmental literature. An advantage, however, of using the CAPE was that these data were collected at school, thus diminishing additional attrition from this study because parents may have been concerned about the additional burden of having their child wear an accelerometer. This concern was seen in a previous study using accelerometry in this school district [41], where permission to wear an accelerometer for 7 days was only provided for 51% of the *n* = 403 kindergarten children who had parental consent to participate in the at-school portion of the study. 

### 4.5. Future Recommendations

There were still children in grades three, four, and five who overestimated (low–high) or underestimated (high–low) their abilities. As evidenced in the academic domain [23,42,43], it is likely that both of these groups will benefit from developing more realistic self-perceptions, but for different reasons. Most urgently, investigation into how to prevent children from underestimating their abilities and ways to help these children recognize their actual skill level is needed. To date, there is limited evidence to support a longitudinal relationship between perceived and actual motor competence (and vice versa), creating a need for more research in this area [4]. It is likely for some underestimating children that this is complex [23] and may require inter- and multi-disciplinary (e.g., psychology, pedagogy, motor development) teams for intervention and investigation. Additionally, further exploration, and confirmation, of the impact overestimating has on motor skill development and engagement in physical activities is also needed. 

Psychological constructs such as perceptions of physical competence develop over time [22,23,24]. Age-related changes in perceptions of competence are established through a combination of cognitive development and socialization experiences. It is believed, however, that socialization experiences contribute heavily to an individual’s personal evaluation of their abilities [23]. For this reason, it is suggested that future research explore the role of different socialization environments (e.g., home, school, community) and how those environments may differentiate how children develop perceptions of their abilities across early and middle childhood and into adolescence. 

## 5. Conclusions

Our findings show that combinations of motor skills and perceptions influence the pattern of engagement or disengagement in active physical recreation across middle childhood. Children with good motor skills and positive physical self-perceptions participated in significantly more active physical recreation than those with low motor skills and low perceptions. Additionally, children with positive perceptions, whether they had strong or poor motor skills, demonstrated positive growth in their motor skills, suggesting positive perceptions are beneficial, even for children with less developed skills. This was a consistent pattern from grade three to grade five. The most concerning participants are those with low perceptions. Children with low perceptions and low motor skills need urgent intervention regarding their motor skills, as they are low, and they know it. Without intervention, these children appear to move toward a path of disengagement in active physical recreation. Further, children with low perceptions, but high motor skills, also need support to recognize their actual skill level as this seemed to have negative consequences on motor competence development. 

## Figures and Tables

**Table 1 ijerph-21-01129-t001:** Score range and interpretation of CAPE active physical recreation scores.

CAPE Dimension (Score Range)	Score Interpretations
Intensity (1–7)	1 = one time in the past four months 2 = two times in the past four months 3 = one time a month 4 = two to three times a month 5 = one time a week 6 = two to three times a week 7 = one time a day or more

**Table 2 ijerph-21-01129-t002:** K-means cluster analysis descriptive statistics and between-group post hoc comparisons.

	Low–Low		High–High		Low FMS–High PPC (L-H)		High FMS–Low PPC (H-L)	
Variable (Range)	Mean	SD	Mean	SD	Mean	SD	Mean	SD
Grade 3								
*n*	31 (boys = 15)		63 (boys = 30)		27 (boys = 12)		34 (boys = 13)	
FMS (0–96) ^a,c,d^	57.71	6.01	69.32	7.32	54.22	6.37	69.74	6.18
PPC (6–24) ^a,b,c,d,e^	12.87	2.69	21.81	1.83	19.15	2.16	15.59	2.66
APR ^a,b,c^	1.37	0.60	1.97	0.66	1.98	0.93	1.98	0.67
Grade 4								
*n*	33 (boys = 14)		46 (boys = 19)		34 (boys = 16)		42 (boys = 21)	
FMS (0–96) ^a,c,d,e^	57.67	6.16	72.89	5.70	58.94	5.15	67.10	3.99
PPC (6–24) ^a,b,c,e^	13.03	2.30	21.46	1.72	20.47	2.03	16.74	1.91
APR ^a,b,c^	1.60	0.82	2.26	0.76	2.12	0.83	2.19	0.77
Grade 5								
*n*	21 (boys = 9)		53 (boys = 24)		40 (boys = 18)		41 (boys = 19)	
FMS (0–96) ^a,b,c,d,e^	55.33	5.54	74.32	5.15	61.03	4.16	69.39	4.18
PPC (6–24) ^a,b,e^	14.52	2.60	21.11	1.74	20.78	2.06	14.76	2.51
APR ^a,b^	1.32	0.57	2.22	0.75	2.19	0.79	1.85	0.87

Note. All differences at *p* < 0.05. ^a^ significant difference between low–low and high–high. ^b^ significant difference between low–low and L-H. ^c^ significant difference between low–low and H-L. ^d^ significant difference between high–high and L-H. ^e^ significant difference between high–high and H-L. FMS = fundamental motor skills; PPC = perceptions of physical competence; APR = active physical recreation.

**Table 3 ijerph-21-01129-t003:** Descriptive statistics of boys’ and girls’ motor skills, perceptions of physical competence, and active physical recreation by cluster group for grades 3, 4, and 5.

	Fundamental Motor Skills					Perceptions of Physical Competence					Active Physical Recreation				
	Boys		Girls			Boys		Girls			Boys		Girls		
Cluster	Mean	SD	Mean	SD	*p*	Mean	SD	Mean	SD	*p*	Mean	SD	Mean	SD	*p*
	Grade 3														
All	67.33	10.79	62.09	7.17	<0.001	19.33	3.77	17.26	4.35	<0.001	1.81	0.68	1.89	0.69	0.348
Low–low	59.07	7.24	56.44	4.46	0.219	14.13	2.62	11.69	2.24	0.001	1.21	0.55	1.51	0.62	0.235
High–high	73.87	6.78	65.18	5.02	<0.001	22.63	1.25	21.06	1.97	0.003	2.01	0.68	1.93	0.65	0.695
Low FMS–high PPC	54.50	7.56	54.00	5.50	0.828	19.67	3.77	18.73	2.52	0.243	1.77	1.03	2.15	0.85	0.273
High FMS–low PPC	73.62	6.70	67.33	4.49	0.003	17.38	1.61	14.48	2.60	<0.001	2.07	0.78	1.92	0.60	0.531
	Grade 4														
All	67.97	8.08	62.59	7.45	<0.001	18.71	3.82	17.72	3.74	0.001	1.97	0.84	2.16	0.81	0.159
Low–low	61.07	6.33	55.16	4.79	<0.001	13.43	2.24	12.74	2.35	0.308	1.39	0.71	1.76	0.88	0.181
High–high	77.00	4.58	70.00	4.54	<0.001	22.16	1.43	20.96	1.77	0.039	2.03	0.77	2.42	0.73	0.101
Low FMS–high PPC	61.37	5.12	56.78	4.22	0.003	21.25	1.92	19.78	1.93	0.027	2.28	0.92	1.99	0.75	0.292
High FMS–low PPC	69.43	3.57	64.76	2.90	0.001	17.19	2.04	16.29	1.71	0.129	2.06	0.77	2.13	0.77	0.300
	Grade 5														
All	69.51	8.25	64.95	7.89	<0.001	19.07	3.67	17.94	3.79	0.004	1.88	0.74	2.09	0.88	0.155
Low–low	58.44	4.00	53.00	5.94	0.004	14.89	2.57	14.25	2.70	0.494	1.25	0.56	1.38	0.60	0.690
High–high	77.46	4.27	71.72	4.34	<0.001	21.75	1.70	20.59	1.62	0.048	1.94	0.66	2.46	0.74	0.014
Low FMS–high PPC	62.61	4.71	59.73	3.21	0.032	21.22	1.83	20.41	2.20	0.228	2.32	0.68	2.09	0.87	0.332
High FMS–low PPC	71.26	3.66	67.77	3.99	0.009	15.63	2.52	14.00	2.29	0.015	1.68	0.73	2.00	0.97	0.174

## Data Availability

Data supporting reported results can be obtained by contacting the contributing author.

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
