# Peer review of "Perceptions Matter! Active Physical Recreation Participation of Children with High and Low Actual and Perceived Physical Competence"

_ijerph, 2024, doi:10.3390/ijerph21091129_

Round 1

Reviewer 1 Report

Comments and Suggestions for Authors

Thank you for the opportunity to review the manuscript.  The authors present a well-organized study with appropriate methodology.  While the results might seem intuitive, the data provides the evidence to support , which elevates the contribution of the study to the scholarship regarding the developing relationship between levels of physical activity and self awareness. The results are informative for practice with those who educate and care for children in this timeframe. 

The ideas are generally well-developed, and literature is appropriate; however, in keeping with the recommendations of the Journal, only two out of 36 are five years or newer (2019). Consequently, I suggest the authors conduct further literature review to ensure that they have not overlooked more recent relative studies.

The authors do not identify a theoretically driven hypothesis in the Introduction. They draw from Stodden et al. work; however, in the Discussion, Harter’s work is introduced to explain the results. I suggest the authors consider introducing Harter’s work in the Introduction. Additionally, a clear definition of middle childhood would be very helpful as this varies by theory, licenses to practice, and in policy work. Clarify the beginning and end of this stage/phase or clarify if you understand human development on a continuum (e.g., Siegler, Robert).

Overall, there are no issues with English; however, I would recommend rewording the first sentence of the abstract to read, “Emerging evidence suggests… “

On line 280, the partial eta squared precedes rather than follows the p value in the first F value report. Please reverse so that it follows the p value.

Finally, within the Future Recommendations, the authors suggest perceptions of one’s self develop over time but stop short of identifying children’s experiences during early childhood that may describe, predict or explain differences in performance and one’s growing sense of self during middle childhood.  I encourage the authors to consider suggesting further research to examine the contribution of early childhood experiences and the role of the family (parenting) and the home environment that may differentiate children as they grow into middle childhood. 

Reviewer 2 Report

Comments and Suggestions for Authors

Thank you for the opportunity to review this manuscript. The manuscript titled "Perceptions Matter! Active Physical Recreation Participation of Children with High and Low Actual and Perceived Physical Competence" is a well-structured study that investigates the differences in active physical recreation participation among children with varying levels of actual and perceived physical competence. The findings offer valuable insights into the role of perceived competence in children's physical activity participation, contributing significantly to the field of physical education and health promotion. Given the importance and relevance of the study, I recommend acceptance with minor revisions to address some formatting and detail-related issues.

Introduction:

The research problem is well-defined, but emphasizing the study's novelty and the specific gaps it addresses in the existing literature would enhance the introduction. The review is comprehensive, but the international context could be elaborated further. Ensure that all citations follow the MDPI format guidelines.

Methods:

The methods section is well organized and detailed. While the K-means clustering method is appropriate, a brief description of its theoretical background would be desirable.

Results 

The data is generally well-presented. However, some tables need better formatting: Ensure that all tables have clear and informative headings and are formatted according to MDPI guidelines.

Discussion:

The discussion is thorough but could deepen into the mechanisms behind the observed effects. Additionally, a more detailed discussion on the study's limitations and future research directions would strengthen this section.

Comments on the Quality of English Language

The language is very fluent and logical, please check again for grammatical errors and other minor issues.
